# An Ancient Mutation in the *TPH1* Gene is Consistent with the Changes in Mammalian Reproductive Rhythm

**DOI:** 10.3390/ijms20236065

**Published:** 2019-12-02

**Authors:** Chenhui Liu, Xunping Jiang, Guiqiong Liu, Teketay Wassie, Shishay Girmay

**Affiliations:** 1Laboratory of Small Ruminant Genetics, Breeding and Reproduction, College of Animal Science and Technology, Huazhong Agricultural University, Wuhan 430070, China; liu890621@webmail.hzau.edu.cn (C.L.); xpjiang@mail.hzau.edu.cn (X.J.); teketay@webmail.hzau.edu.cn (T.W.); shishay@webmail.hzau.edu.cn (S.G.); 2Key Laboratory of Agricultural Animal Genetics, Breeding and Reproduction of the Ministry of Education, Wuhan 430070, China

**Keywords:** melatonin-related genes, positive selection, functional diversification, reproductive rhythm

## Abstract

The reproductive rhythm undergoes several changes during the evolution of mammals to adapt to local environmental changes. Although the critical roles of melatonin (MLT) in the formation of reproductive rhythm have been well established, the genetic basis for the changes of reproductive rhythm remains uncertain. Here, we constructed the phylogenetic trees of 13 melatonin synthesis, metabolism and receptor genes, estimated their divergence times, and calculated their selection pressures. Then, we evaluated the effect of positively selected and functionally related mutations on protein activity. Our results showed that there were significant positive selection sites in the three major genes, including tryptophan hydroxylase 1 (*TPH1*), tryptophan hydroxylase 2 (*TPH2*) and indoleamine-2,3-dioxygenase 1 (*IDO1*) that are involved in melatonin synthesis, metabolism and function. At the protein level, amino acids at the 442nd site of TPH1 protein and the 194th, 286th, 315th and 404th sites of IDO1 protein were under positive selection, and the variants of the amino acid in these sites might lead to the changes in protein function. Remarkably, the 442nd site of these positive selection sites is in the tetramerization domain of TPH1 protein, and it is proline or leucine. At this site, 89.5% of the amino acid of non-seasonal reproducing mammals was proline, while that of 88.9% of seasonal reproducing mammals was leucine. This variation of the amino acid was derived from the T/C polymorphism at the 1325th site of the *TPH1* gene coding sequence, which significantly altered the TPH1 activity (*p* < 0.01). Interestingly, the predicted age of the allele C in the mammalian genome appeared about 126.6 million years ago, and allele T appeared about 212.6 million years ago, indicating that the evolution of the *TPH1* gene was affected by the two mammalian split events and the K-T extinction event. In conclusion, the T/C polymorphism at the 1325th site in the *TPH1* gene coding sequence altered TPH1 activity, suggesting that this polymorphism is consistent with the reproductive rhythm of mammals.

## 1. Introduction

Seasonal reproduction is a strategy for mammals to adapt to environmental changes. It allows mammals to reproduce in the most favorable period of the year for the survival and growth of their offspring. The optical signal is a significant factor in controlling the seasonal reproduction of animals [1]. Melatonin (MLT), secreted by the pineal gland, is the key to synchronizing external optical signals and neuroendocrine signals [2].

Melatonin is synthesized and secreted by the pineal gland, retina, salivary gland, thymus, thyroid, kidney and liver [3], of which the pineal gland is the most important synthetic and secretory organ. There are four steps in the melatonin synthesis pathway in vivo. Firstly, tryptophan is converted to 5-hydroxytryptamine (5-HTP) by tryptophan hydroxylase (TPH) and decarboxylated by aromatic-L-amino acid decarboxylase (AAAD) to produce 5-hydroxytryptamine (5-HT), which is also known as serotonin. Then, serotonin is acetylated with aromatic alkyl-N-acetyltransferase (AANAT) and methylated with hydroxyl indole-o-methyl transferase (HIOMT) to form melatonin. Among these, TPH is the rate-limiting enzyme in serotonin synthesis [4]. AANAT and HIOMT are the rate-limiting enzymes in melatonin synthesis [5]. Notably, there are two *TPH* genes in the mammalian genome, including *TPH1* and *TPH2* [6]. The expression products of the two genes have similar structures and functional domains, and thus their functions are similar [7]. However, both genes were expressed in different organs: the *TPH1* is mainly expressed in the pineal gland, retina, kidney, lung, kidney, duodenum, liver and adrenal gland, while the *TPH2* is mainly expressed in the brain stem [7,8].

Melatonin is released into the cell membrane and binds to specific receptors to exert hormone function [9]. There are two melatonin receptors in mammals, namely melatonin receptor 1A (MTNR1A) and melatonin receptor 1B (MTNR1B). These two receptors are the members of G protein-coupled receptors, and MTNR1A is more common than MTNR1B in mammals [4].

The metabolic processes of melatonin are widely found in various tissues, such as the liver, plasma, lung, spleen, heart, adrenal gland and brain, among which the liver and brain are the main ones [10]. Melatonin is mainly converted to 6-hydroxymelatonin (6-HMT) and N-acetylserotonin (NAS) by cytochrome P450, which is further metabolized to form sulfate or glucoside compounds and then excreted through urine. It is documented that four cytochrome P450 proteins, including CYP1A1, CYP1A2, CYP1B1 and CYP2C19, are involved in the metabolism of melatonin, of which CYP1A1 and CYP1A2 are the main enzymes involved in melatonin metabolism in the liver, and the metabolite is 6-HMT. At the same time, a small amount of melatonin was metabolized by CYP1A2 and CYP2C19, and the metabolite was NAS [11]. In other tissues, especially in the intestine and cerebral cortex, melatonin metabolism is mainly catalyzed by CYP1B1 and its metabolite is 6-HMT [12]. In the brain, melatonin can be catalyzed by indoleamine-2,3-dioxygenase (IDO) to form N^1^-acetyl-N^2^-formyl-5-methoxyknuramine (AFMK), which is then converted to a more stable *N*-acetyl-5-methoxyknuramine (AMK) [13] by arylamide–formylase. There are two IDO proteins in mammals, namely IDO1 and IDO2, whose structures, functions, and expression regions are similar in vivo, but the expression process is regulated by different cytokines [14].

Variations in the sequences and expression of melatonin-related genes are strictly related to the human mood [15], behavior [16] and diseases [17]. Among these genes, polymorphisms of the *MTNR1A* gene encoding melatonin receptor are associated with the reproductive rhythm in sheep [18]. However, the association between genetic variations of other melatonin-related genes and mammalian reproductive rhythms remains unclear. Therefore, the purpose of this study was to detect: the allelic variations associated with the reproductive rhythm in the above melatonin-related genes and their evolutionary history.

## 2. Results

### 2.1. Evolution of Melatonin-Related Genes

The positive selection analysis of the 13 melatonin-related genes is shown in Appendix A. The *ω* values changed significantly among the codons of these genes as the likelihood ratio test (LRT) values of the M0–M3 model are significant in the Chi-square test (*p* < 0.05). Eight of these 13 melatonin-related genes have positively selected sites because their LRT values in the M7–M8 model are significant (*p* < 0.05). Further Bayesian Empirical Bayesian (*BEB)* analysis showed that *TPH1*, *TPH2* and *IDO1* of the eight genes had significant positive selection sites (*p* < 0.05).

The positive selection analysis of *TPH1*, *TPH2*, and *IDO1* genes are shown in Table 1. For the *TPH1* gene, the LRT value of the M7–M8 model was 36.742 (*p* < 0.01), and 1.8% of *TPH1* codons were positively selected. However, the LRT value of the M1a–M2a model was 0 (*p* > 0.05). The *BEB* analysis showed that there were two significant positive selection sites (*p* < 0.01) at the protein level, which are the 97th and 442nd sites.

The results of the *TPH2* gene were similar to that of the *TPH1* gene. The LRT value of M7–M8 model was 47.952 (*p* < 0.01), and 1.7% of the codons were positively selected. However, the LRT value of the M1a–M2a model was 0 (*p* > 0.05). Besides, the *BEB* analysis exhibited two significant positive selection sites at protein level (*p* < 0.01), 47th and 66th sites.

For the *IDO1* gene, the LRT values of M1a–M2a and M7–M8 models were 23.524 and 52.290, respectively (*p* < 0.01). The M1a-M2a model showed that 0.4% of the codons were positively selected, while 25% of the codons were positively selected in the M7–M8 model. The *BEB* analysis in this model found ten significant positive selection sites at the protein level (*p* < 0.05): they are the 194th, 197th, 205th, 219th, 229th, 286th, 315th, 336th, 400th and 404th sites.

The adaptive branching-site random effect likelihood (aBSREL) model was used to test the positively selected branches of *TPH1*, *TPH2* and *IDO1* genes, respectively, and the results are shown in Figure 1. The *TPH1* gene was significantly positive selected on the horse and the ancestor of Green monkey and Sumatran orangutan (*p* < 0.05), with 3.6% and 1.2% of codons positively selected, respectively. The *TPH2* gene showed significant positive selection on golden hamster and crab-eating macaque (*p* < 0.05), and 0.76% and 0.44% of the codons of the expression products were positively selected in these two species, respectively. The *IDO1* gene had a significant positive selection on the ancient Boreoeutheria (*p* < 0.01), and 17% of the codons in this branch were positively selected.

### 2.2. Functional Divergence of Melatonin-Related Proteins

The functional diversification analysis examined whether the functions of subtype proteins diverged, including TPH1 and TPH2, IDO1 and IDO2, and four cytochrome P450 isoforms, respectively. Then, it was determined which amino acid variants caused their functional divergence (Table 2). In 28 mammals, the LRT values of TPH1/TPH2, IDO1/IDO2 were 42.837 and 91.682 (*p* < 0.01), respectively, indicating that the structure and function of TPH1 and TPH2, IDO1 and IDO2 had visible divergence in these mammals. For the six combinations of the four cytochrome P450 protein isoforms, the LRT value of CYP1A1/CYP1A2 was 0.176 (*p* > 0.05), indicating that the structure and function of CYP1A1 and CYP1A2 proteins did not diverge significantly in these mammals, while the protein divergence of the other five combinations was extremely significant (*p* < 0.01).

The results in Appendix A indicated that the 442nd site of TPH1, which was positively selected, might be related to the functional divergence of TPH1 and TPH2. For IDO1, four positively selected sites—the 194th, 286th, 315th and 404th—might be related to the functional divergence of IDO1 and IDO2.

### 2.3. The T/C Polymorphism at the 1325th Site in TPH1 Gene Change Enzyme Activity

There is a total of five amino acid sites on the TPH1 and IDO1 proteins that are both positively selected and related to functional divergence. The 442nd site of TPH1 is located in the tetramer-binding domain (Figure 2B), and three of the four positive selected sites of IDO1 are in the pyridoxal phosphate-dependent transferase domain, including the 194th, 286th and 315th sites. There was no polymorphism at the 442nd site of TPH1 within the species included in the Ensemble database. For IDO1, two synonymous variants at the 315th and 404th sites can be detected in human and house mouse, respectively, and the 194th and 286th sites showed no polymorphism within the species included in the database. Therefore, we aligned the amino acid sequences of 28 mammals and found that it was leucine (L) or proline (P) at 442nd site of TPH1, which was skewed distributed in seasonal or non-seasonal reproducing mammals (Figure 2A). The amino acid at the 442nd site was proline in 89.5% of non-seasonal reproducing mammals, while it was leucine in 88.9% of seasonal reproducing mammals.

The nucleotide sequence alignment of the *TPH1* gene in 28 mammals showed that the variation of the 442nd amino acid originated from the T/C mutation of the 1325th nucleotide site in the coding sequence. The allelic distribution of this site was consistent with the distribution of the amino acid in seasonal or non-seasonal reproducing mammals. The allele C was in 89.5% of non-seasonal reproducing mammals, while allele T was in 88.9% of seasonal reproducing mammals. The allele C appeared in bats 83.0 million years ago (mya) and spread in rodents 52.4 mya. The allele T appeared 212.6 mya and spread in primates 25.1 mya.

The enzyme assay of TPH1-1325T and TPH1-1325C proteins revealed that the T/C polymorphism of the 1325th site could significantly change the activity of TPH1 protein (Figure 2C). The activity of TPH1-1325T protein was 0.020 U/mg and that of TPH1-1325C was 0.004 U/mg. The activity of TPH1-1325T was five times higher than that of TPH1-1325C, and their difference was highly significant (*p* < 0.01).

## 3. Discussion

### 3.1. TPH1, TPH2 and IDO1 Genes are Positively Selected in Mammals

Calculating the ratio of the number of non-synonymous substitution sites to that of synonymous substitution sites is a standard method for the positive selection test. However, this method has been criticized for producing excessive false-positives and for being sensitive to various disturbing factors [19].

Eight of the 13 melatonin-related genes studied here had a significant difference in *LRT* values (*p* < 0.05) in the M7–M8 model. As for the eight genes, only the *IDO1* gene had different *LRT* values in both M1a–M2a and M7–M8 models (*p* < 0.05). For the other seven genes, including *TPH1*, *TPH2*, *DDC*, *MTNR1B*, *IDO2*, *CYP1A1* and *CYP1B1* genes, the *LRT* values of the M1a–M2a model were not significantly different (*p* > 0.05) while those of the M7–M8 model showed significant differences (*p* < 0.05). Two of these seven genes, *TPH1* and *TPH2*, had significant positive selection sites (*p* < 0.05) in *BEB* analysis. The study of Twyman showed similar results in detecting the selection pressure of the *CYP2J19* gene [20]. Therefore, we accepted the results of M7–M8 that positive selection sites existed in these seven genes.

Berlin and Smith pointed out that the M7–M8 model may overestimate non-positive selection sites as positive selection sites, leading to false-positive results [21]. To avoid misjudging false-positive sites, we used the aBSREL branch length-site model to detect the selection of *TPH1* and *TPH2* gene. The M7–M8 model in the codeml program assumes the *ω* values in different branches are fixed. However, the aBSREL program assumes that the *ω* values are variable between codons and in different branches. Therefore, the model can detect not only positive selection sites but also positive selection branches during evolution [22], thus complementing the results of codeml. The results of the aBSREL showed that the *TPH1* gene was significantly positively selected in two branches of the phylogenetic tree (i.e., the horse branch and the ancient branch of the green monkey-Sumatran orangutan). The *TPH2* gene was positively selected in the branches of golden hamster and crab-eating macaque, respectively. The results were consistent with those of the M7–M8 model in the codeml program, indicating that our results of *TPH1* and *TPH2* genes were credible. Although, the gene has multiple transcript variants in a specific species in GenBank, we only chose one sequence which is most similar to the human sequence for research. Obviously, this method will actually lose some polymorphic information, which will lead to a reduction in the accuracy of selection pressure on different branches. Therefore, the more accurate selection pressure of each branch needs to be further studied in specific species.

### 3.2. Effect of T1325C Polymorphism of the TPH1 Gene on Seasonal Reproduction in Mammals

In mammals, the expression tissues of the *TPH1* gene and the synthesis tissues of melatonin are similar. Even if the expression product of the *TPH1* gene was not regarded as a rate-limiting enzyme in the melatonin synthesis pathway [23], recent studies still believed that it could regulate the synthesis of serotonin and melatonin in the pineal gland and peripheral tissues [6,24].

Positive selection and functional divergence analysis showed that 442L was not only a significant positive selection site but also related to functional divergence. This site was in the tetramer region of TPH1 protein, which is an alpha-helix structure at the C-terminal. In seasonal reproducing mammals, the amino acid at 442nd site was mainly leucine (88.9%), while the amino acid was mainly proline (89.5%) in non-seasonal reproducing mammals. However, the amino acid here was proline in Weddell’s seal, which is a seasonal reproducing mammal—probably because Weddell’s seal is an aquatic mammal.

Comparing the nucleotide sequences and protein sequences, we found that the variation of TPH1 protein L442P originated from the T1325C mutation of the *TPH1* gene. When the T1325C site of the *TPH1* gene mutated from T to C, the 442nd amino acid changed from L to P, thus reducing the number of leucine in the tetramer region. Our results in the activity assay showed a higher activity of TPH1-1325T (four leucine in the tetramer region) and lower activity of TPH1-1325C (three leucine in the tetramer region). The four leucine in this region formed a leucine zipper. The leucine zipper maintained TPH1 tetramers that are stable and active. The decrease of the leucine amount results in dimmers or monomers of the TPH1 that are unstable or inactive [25]. The lower active of TPH1 with 442P in our study supports Marshall’s study.

TPH1 protein catalyzes tryptophan to produce serotonin, which provides a substrate for the synthesis of melatonin. Whether in the retina or pineal, the expression of the *TPH1* gene showed a trend of low during the day and high during the night [26], which was consistent with the rule of melatonin synthesis [27]. However, serotonin in vivo is the highest at midnight and noon and the lowest at dawn and dusk [28]. We suppose that serotonin may be synthesized before melatonin as a substrate at night. Therefore, the increased activity of TPH1 can advance the nighttime peak of melatonin by improving the synthesis efficiency of serotonin, leading to the increased sensitivity of animals to diurnal changes, which provides a basis for seasonal reproduction. On the contrary, the decrease of TPH1 activity may lead to a decline in melatonin synthesis efficiency. Thus, the sensitivity of animals to optical signals would be reduced and eventually lead to the weakening or disappearance of seasonal reproduction of mammals.

However, there are exceptions to the correlation of the 1325th site of the TPH1 gene coding sequence to the seasonal reproduction of mammals. For example, in the *TPH1* gene of gray short-tailed opossum and chimpanzee, the allele at the T1325C site is T, but these two species were not seasonal breeders. It is inferred that this may be due to the complex process of seasonal reproduction controlled by multiple genes. The effect of T1325C mutation on melatonin synthesis in these species may be offset by mutations in other genes that are also involved in this pathway, especially AANAT and HIOMT speed-limiting enzymes.

### 3.3. Seasonal Reproduction Improved the Adaptability of Mammals to Cretaceous–Tertiary Extinction Events

The estimation of the divergence time of the *TPH1* gene in 28 mammals showed that the allele T at the 1325th site has existed for over 210 million years, and allele C probably first appeared about 83.0 million years ago (mya). We mapped the TPH1 evolutionary history from 210 mya to the present in Figure 3.

The platypus is the oldest and the most ancient mammal that still exists today; it appeared at the end and the warmest time of the Cretaceous (145.0–165.0 mya) [29]. After millions of years, it has neither gone extinct nor evolved significantly. The platypus is a seasonal reproducing mammal. The base at 1325th site of the *TPH1* coding sequence is T in the platypus, suggesting the base at this site in mammals may be T at first and the T base may control seasonal reproduction in platypus. The base at 1325th site changed from T to C and spread in ancestors of rodents (mainly non-seasonal reproduction animals) about 83.0 mya (Figure 2A), which coincided with the time of the first major split events in mammals (100.0–85.0 mya) [30]. During this period, the first split event of mammalian species occurred, and the ancestors of many existing species appeared, such as hamsters, rabbits, goats and sheep (all about 96.0 mya). The disappearance of seasonal reproductive characteristics is conducive to more offspring and a rapid expansion of population size.

Later, allele T spread again about 70.9 mya, which was close to the time of Cretaceous–Tertiary (K-T) extinction (about 65.5 million years ago) [31]. The K-T extinction event was a mass extinction event in the history of the Earth which led to the extinction of most of the animals and plants at that time and is famous due to the rise of mammals [31]. The most likely cause of this extinction is multiple impact events [32] or long-term volcanic eruptions [33], which threw a large amount of dust into the atmosphere. The dust blocked the sunlight, causing the temperature to drop sharply [29], and resulting in large-scale plant deaths due to reduced photosynthesis, thus resulting in a shortage of animal food sources, leading to the extinction of a variety of animals. We hypothesize that seasonal reproduction improves the adaptability of some mammals to this extreme environmental pressure. For non-seasonal reproducing mammals, such as rodent ancestors, their small size and cave-dwelling characteristics helped them to survive and reproduce with less food, and it was easy to find suitable habitats for population continuation. In the genome of these mammals, allele C is retained. For seasonal reproducing mammals with the allele T at the 1325th site, such as the ancestors of existing cattle, horse, goat and sheep, they can change their reproductive rhythm by sensing light changes and produce offspring when food is relatively adequate, thus ensuring the survival of offspring and population continuity. Seasonal reproduction improved their adaptability to the surging environmental pressure, making individuals with allele T dominant in natural selection.

After the K-T extinction, the Earth entered the Paleocene (65.0–55.0 mya). During this period, the light returned to normal, global temperature increased by 5–8 °C, and the Paleocene–Eocene Thermal Maximum (PETM or IETM) occurred. As a result, environmental pressure relaxed [34]. Under this background, a second mammalian split event occurred (45.0–25.0 mya). Non-seasonal reproduction enabled mammals to have more offspring to gain an advantage in competing for the ecological space once occupied by extinct species [35]. In this context, the individuals with allele C predominate in natural selection. The second spread of allele C in the mammalian genome occurs.

We inferred that the evolution of the *TPH1* gene in mammals might be influenced by two mammalian split events and the K-T extinction event. The T1325C site under positive selection could be evolved under temperature-associated selection. The environmental changes play a dominant role in the spread of the *TPH1* T1325C site and the change of seasonal reproduction in mammals. When light and temperature decreased, seasonal reproduction could improve the adaptability of populations to extreme environmental pressures, which caused allele C to spread. Once light increased and the climate warmed up, non-seasonal reproducing populations could gain more offspring under relaxed environmental pressures. Thus, individuals with allele T predominated in natural selection.

We mapped the mammalian evolutionary history from 160 million years ago (mya) to the present. Five periods were mainly experienced: Jurassic (200.0–145.0 mya), Cretaceous (145.0–65.0 mya), Paleogene (65.0–23.3 mya), Neogene (23.3–2.6 mya) and Quaternary (2.6 mya–now). During these periods, the Earth was mainly in the Triassic–Tertiary interglacial period (200.0–2.0 mya), with a warm climate. However, about 65 million years ago, there was a cold period due to multiple impacts (with a maximum radius of about 150 km) and long-term volcanic eruptions. After this intense period, the global temperature rose rapidly. During the Quaternary, the Earth entered the Quaternary Ice Age, the second cold period occurred, and the temperature decreased sharply.

During these 160 million years, mammals experienced two splits and two extinctions. The first split occurred about 100–85 mya, which was the warmest time in the whole Cretaceous. Subsequently, the K-T extinction happened about 65.5 mya. After the K-T extinction, mammals entered the second major split (about 40–25 million years ago). The second extinction occurred during the Quaternary Ice Age, during which many mammals were extinct.

The evolutionary history of the *TPH1* gene is highly consistent with the change of reproductive rhythm in mammals. Based on the phylogenetic tree of 29 species, with the platypus (red, seasonal reproduction) as a model, the three changes of reproductive rhythm and the 1325th site of *TPH1* gene were found at the same time in bat (gray, non-seasonal reproduction), European shrew (red, seasonal reproduction) and marmoset (gray, non-seasonal reproduction), respectively. Therefore, these four species (arrange from top to bottom) were chosen to describe the evolution of the *TPH1* gene in mammals.

## 4. Materials and Methods

### 4.1. Data Collection

A total of 259 coding sequences for 13 melatonin-related genes from 29 species were collected from the NCBI database. Of the 29 species, 28 of them are mammals, and one is zebrafish as an outgroup. The Latin and English names of the species and the genes with accession numbers are shown in Appendix A. If there are multiple transcript variants of a gene in a specific species, only one sequence which is most similar to the human sequence will be selected for further research.

### 4.2. Phylogenetic Trees and Divergence Time

To construct the evolutionary history of melatonin-related genes, the coding sequences of 13 genes were aligned with MEGA-X [36] and their phylogenetic trees were constructed by maximum likelihood method (ML) with 1000 bootstrap values [37]. Pairwise deletion was used to delete the sites with less than 95% coverage among species. The relative substitution rates of nucleotides were calculated by Kimura’s two-parameter model [38]. The EvolView online program (http://www.evolgenius.info/evolview/) was used to modify the phylogenetic trees [39].

The maximum likelihood method and Kimura two-parameter model were used to infer the ancestral sequences for these TPH proteins in 29 species [40]. The divergence times of 13 genes were calculated based on the divergence time [41,42,43] of the human and gray short-tailed opossum (155.0–166.0 million years ago), and human and gorilla (5.1–11.8 million years ago) by the clocks process based on the RelTime method in MEGA X [44].

### 4.3. Positive Selection Assessment

Three pairs of nested models (M0–M3, M1a–M2a, and M7–M8) were employed in the site model of the codeml program in the Phylogenetic analysis by maximum likelihood (PAML) package to predict the potential positively selected sites in melatonin-related genes. The selection pressure of genes was indicated by the ratio (*ω*) of non-synonymous substitution and synonymous substitution number (*d_N_*/*d_S_*). When *ω* = 1, the gene is neutrally selected; when *ω* < 1, the gene is purified; when *ω* > 1, the gene is positively selected [45].

In three pairs of nested models (M0–M3, M1a–M2a, M7–M8), the M0 model assumes that all loci have the same *ω* values, while the M3 model assumes that the *ω* values of each codon are independent and have three possibilities: more than 1, equal to 1, and less than 1. The M0–M3 model is not used for the positive selection test, but for detecting whether the *ω* values between loci are consistent or not. Both M1a–M2a and M7–M8 models consisted of null hypothesis models (M1a and M7), which assume that all codons are under near-neutral selection, and alternative hypothesis models (M2a and M8) assume that positive selection exists. The difference between the two pairs of models was that the M7–M8 models used *β* distribution to fit the value of *ω* [46]. Compared with the M7–M8 model, the M1a–M2a model is more stringent [46], while the M7–M8 model is more sensitive and can detect weaker positive selection [21].

For each pair of models, the likelihood ratio test (LRT) was used to evaluate the fitting degree of nested models with the likelihood logarithm (ln*L*) of the paired model. Significance was evaluated by the chi-square test with 4, 2 and 2 degrees of freedom for M0–M3, M1a–M2a and M7–M8 models, respectively. Naive Bayesian (NEB) and Bayesian Empirical Bayesian (*BEB*) methods in M2a and M8 were used to calculate the posterior probability of each site selected by codons and to judge the site selection from the significant test results. According to the suggestions given in the codeml instructions, we ignored *NEB* and used *BEB* results directly [47].

An adaptive branching-site random effect likelihood model (aBSREL) was used to calculate the selection pressure (*ω*) on the codons of melatonin-related genes in various branches of phylogenetic trees. In this model, branches are divided into foreground branches (branches to be detected) and background branches in advance. A likelihood ratio test was used to evaluate the fitting degree of foreground branches to near-neutral selection (null hypothesis) and positive selection (alternative hypothesis) and then to determine whether positive selection has taken place [22]. The calculation program of the model is provided by the DataMonkey server (http://www.datamonkey.org/).

### 4.4. Functional Diversification Analysis

The structural divergence of melatonin-related proteins between different branches of the phylogenetic tree was examined, and their amino acid sites related to functional divergence were predicted with the Gu99 and Type-I functional divergence in the Diverge v.3.0 program. After comparing the amino acid sequences of all proteins, the structural divergence of proteins between different branches was calculated by the likelihood ratio test (LRT), and their functional divergence was inferred. Type-I functional divergence (site-specific rate shift) refers to the evolutionary process resulting in site-specific rate shifts after gene duplication. It identifies amino acid residues highly conserved in one gene copy and highly variable in the other. The probability of a residue being under Type-I divergence is denoted as *θ*. *Q_(k)_* is the site (*k*)-specific score corresponding to the posterior probability that site k is related to type-I functional divergence. Based on Thompson and his colleagues’ report, 0.9 was chosen as the critical value of *Q_(k)_*, and the site of *Q_(k)_* > 0.9 was regarded as an important amino acid leading to functional divergence [45].

The polymorphism of sites which are the positive selected and related to functional divergence was detected with the variant tables (Appendix A) of the Ensemble database.

### 4.5. Structural Analysis and Homology Modeling

The InterPro server (http://www.ebi.ac.uk/interpro/scan.html) was used to predict the conservative domain of melatonin-related proteins [48]. Three-dimensional structures of melatonin-related proteins were predicted with the I-TASSER (http://zhanglab.ccmb.med.umich.edu/I-TASSER/) online server [49]. The PyMOL v1.6.x (San Carlos, California, USA) was used to visualize the 3D structure as well as the positioning of conservative domains and positively selected sites.

### 4.6. Cell Expression and Enzyme Activity Determinations

The effect of sites that were positively selected and the functional divergence associated with enzyme activity were examined. The wild-type *TPH1* gene was synthesized based on the coding sequence of sheep *TPH1* gene (NC_019459.1) in NCBI, and its expression product was named as TPH1-1325T. Then, the coding sequence of the T1325C mutated *TPH1* gene was obtained by replacing the 1325th nucleotide on wild-type *TPH1* gene with cytosine from thymine, and the expressed product was named as TPH1-1325C. These two sequences were artificially synthesized and subcloned into pVAX1-asd vector by adding *EcoRI* and *NotI* digestion sites at 5′and 3′ ends, respectively. Subsequently, DNA sequencing was used to detect the accuracy of the two inserts.

To express the protein from the cloned sequences, HEK293T cells were cultured at 37 °C with 5% CO_2_ and 95% humidity for 24 h in Dulbecco’s modified Eagle medium (DMEM, high glucose 4.5 g/L, Gibco, Invitrogen) supplemented with 10% heat-inactivated fetal bovine serum (FBS, Gibco), and 1% l-glutamine, 0.5% penicillin-streptomycin. Then, cells were transfected with a lipofectamine 2000 (Invitrogen) transfection kit. After 48 h of incubation, the total proteins were extracted with the Tryptophan Hydroxylase assay kit (Haling biological technology CO., LTD., Shanghai, China) according to the instructions. The concentration of total protein was determined by the BCA protein assay kit, and the relative concentration of TPH protein was determined by the Western blot. Finally, according to the steps provided by the tryptophan hydroxylase assay kit instructions, the two TPH1 proteins of equal amounts were added to the reaction solution, and their enzyme activities were measured at 750 nm.

## 5. Conclusions

It is concluded that the T/C polymorphism at 1325th site in the *TPH1* gene coding sequence changed the TPH1 enzyme activity and that this polymorphism is consistent with the reproductive rhythm of mammals.

## Figures and Tables

**Figure 1 ijms-20-06065-f001:**
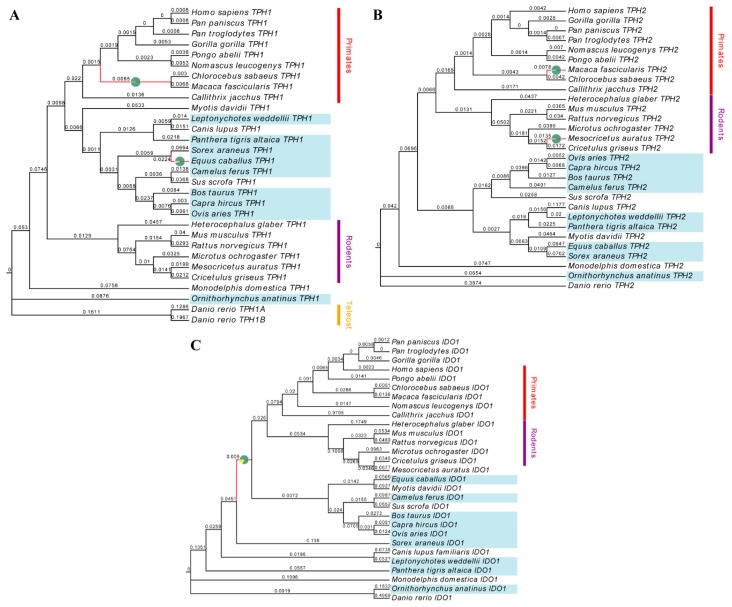
Phylogenetic trees of *TPH1*, *TPH2* and *IDO1* genes. The phylogenetic trees of *TPH1*, *TPH2* and *IDO1* genes are shown in (**A**), (**B**) and (**C**), respectively. The number on each branch represents the relative length of the branch. Colored bars and text describe the brief classification of species. The species with the blue background are seasonal breeders, while those without background are annual breeders or are not well defined. The red branches show that these lineages have a significant positive selection (*p* < 0.05). The pies of red branches show the proportion of positive selection sites (the yellow part) and non-positive selection sites (the green part) over the entire sites.

**Figure 2 ijms-20-06065-f002:**
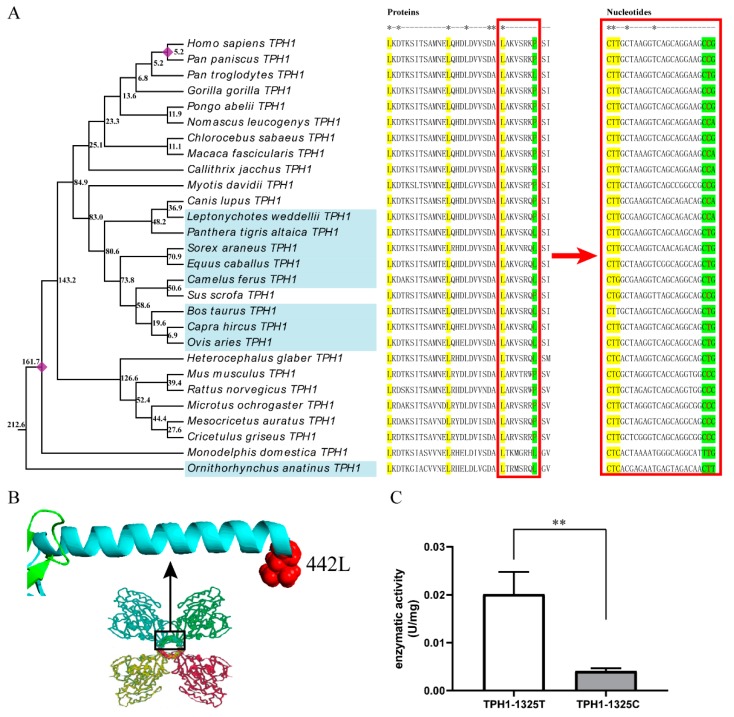
T1325C polymorphism of the *TPH1* gene and its effect on enzyme activity. (**A**) The variation schematic map of *TPH1* genes in mammalian evolution. The phylogenetic tree of *TPH1* genes is shown on the left. The species with the blue background are seasonal breeders, while those without background are annual breeders or are not well defined. The number of nodes indicated the divergence time of the *TPH1* gene. The nodes labeled with a purple diamond are two calibration constraints. The amino acid and nucleotide sequences of the tetramerization domains of these species are shown on the right. There are four leucine in the tetramerization domain. The first three leucine have no difference among species, and the amino acids and corresponding nucleotides are labeled with the yellow background. The last leucine is at the 442nd site, and the corresponding nucleotides are colored with green. The 1325th site in nucleotide sequences was labeled with red. (**B**) The partial three-dimensional structure, positive selection sites and tetramer of TPH1 proteins. The red residue represented the 442nd site. The tetramerization domain is an alpha-helix at the C-terminal of protein and marked in cyan. The picture below the structure of the alpha-helix indicates the tetramer of TPH1. Each monomer is in a different color. The tetramer is anchored by the C-terminal tetramerization domain and additional inter-subunit hydrophobic interactions between the regulatory domains. (**C**) Enzymatic activity of TPH1-1325T and TPH1-1325C proteins. ** indicates significant differences (*p* < 0.01).

**Figure 3 ijms-20-06065-f003:**
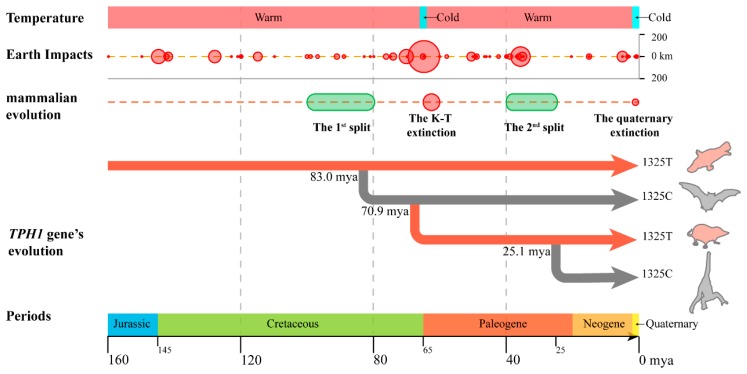
The scheme of *TPH1* gene evolution in mammals.

**Table 1 ijms-20-06065-t001:** Likelihood values and parameter estimates for tryptophan hydroxylase (TPH) genes in site models.

Gene	Model ^a^	ln *L*	Estimates of Parameters ^a^	LRT ^b^	Positive Sites ^c^
*TPH1*	M2a	−6828.237	*p*_1_ = 0.953, *p*_2_ = 0.047, *p*_3_ = 0.000	0.000	97, 442
			*ω*_1_ = 0.075, *ω*_2_ = 1.000, *ω*_3_ = 9.527		
	M1a	−6828.237	*p*_1_ = 0.953, *p*_2_ = 0.047		
			*ω*_1_ = 0.075, *ω*_2_ = 1.000		
	M8	−6817.136	*p*_0_ = 0.982, *p* = 0.472, *q* = 4.095	36.742 *	
			(*p*_1_ = 0.018), *ω* = 1.391		
	M7	−6835.507	*P* = 0.348, *q* =2.304		
*TPH2*	M2a	−8015.897	*p*_1_ = 0.949, *p*_2_ = 0.043, *p*_3_ = 0.008	0.000	47, 66
			*ω*_1_ = 0.034, *ω*_2_ = 1.000, *ω*_3_ = 1.000		
	M1a	−8015.897	*p*_1_ = 0.949, *p*_2_ = 0.051		
			*ω*_1_ = 0.034 *ω*_2_ = 1.000		
	M8	−7982.384	*p*_0_ = 0.983, *p* = 0.044, *q* = 0.337	47.952 *	
			(*p*_1_ = 0.017), *ω* = 1.228		
	M7	−8006.360	*p* = 0.109, *q* = 0.929		
*IDO1*	M2a	−8015.897	*p*_1_ = 0.666, *p*_2_ = 0.293, *p*_3_ = 0.004	23.524 *	194, 197, 205, 219, 229, 286, 315, 336, 400, 404
			*ω*_1_ = 0.125, *ω*_2_ = 1.000, *ω*_3_ = 2.447		
	M1a	−8015.897	*p*_1_ = 0.673, *p*_2_ = 0.327		
			*ω*_1_ = 0.120, *ω*_2_ = 1.000		
	M8	−7982.384	*p*_0_ = 0.750, *p* = 0.816, *q* = 3.690	52.290 *	
			(*p*_1_ = 0.250), *ω* = 1.204		
	M7	−8006.360	*P* = 0.078, *q* = 0.098		

^a^ Two pairs of nested models (M2a–M1a; M8–M7) were showed. In M1a and M2a, *p*_1_, *p*_2_ and *p*_3_ were the proportions of sites with 0 < *ω* < 1, *ω* = 1 and *ω* > 1. In M7 and M8, *p* and *q* were the β distribution parameters, and *p*_0_ and *p*_1_ were the proportions of sites with 0 < *ω* ≤ 1 and *ω* > 1. Sequences were translated from nucleotides to amino acids. The F3×4 CodonFreq model tested the average nucleotide frequencies at three codon positions.; ^b^ Likelihood ratio test (LRT) values were followed the chi-square distribution with 4 (M3 and M0), 2 (M2a and M1a) and 2 (M8 and M7) degrees of freedom, respectively. * indicates *p* < 0.05.; ^c^ Amino acid residues were determined with sheep protein sequences. Only significant positive selection sites were listed.

**Table 2 ijms-20-06065-t002:** Functional divergence estimated in melatonin-related genes.

Comparison (Cluster 1/Cluster 2)	*θ* ^a^	SE ^b^	LRT ^c^	*N* (0.9) ^d^
TPH1/TPH2	0.513	0.061	42.837 *	5
IDO1/IDO2	0.560	0.080	91.682 *	16
MTNR1A/MTNR1B	0.554	0.070	45.038 *	5
CYP1A1/CYP1A2	−0.019	0.093	0.176	0
CYP1A1/CYP1B1	0.802	0.102	60.455 *	16
CYP1A1/CYP2C19	0.660	0.116	34.390 *	4
CYP1A2/CYP1B1	0.766	0.100	62.899 *	13
CYP1A2/CYP2C19	0.892	0.118	57.428 *	22
CYP1B1/CYP2C19	0.839	0.100	77.931 *	51

^a^*θ*: the coefficient of functional divergence.; ^b^ SE: standard error.; ^c^ LRT is a likelihood ratio test. The value approximately follows a chi-square distribution with one degree of freedom. * indicated *p* < 0.05.; ^d^
*N*
_(0.9)_ means the numbers of divergent residues when the cut-off value *Q_(k)_* is over 0.9. The specific amino acids related to protein functional divergence are listed in Appendix A.

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
