# Peer review of "An Ancient Mutation in the TPH1 Gene is Consistent with the Changes in Mammalian Reproductive Rhythm"

_ijms, 2019, doi:10.3390/ijms20236065_

Round 1

Reviewer 1 Report

Review report on Manuscript 611709

An Ancient Mutations in the TPH1 Gene is Consistent with the Changes

in Mammalian Reproductive Rhythm

Scope of the article is hot topic in recent time. Key role of melatonin on seasonal reproductive processes has a long history, however exact time point of its switch on in this regulation is not clear till now.  Melatonin synthesis has several key elements (enzymes), which could be changed during the evolutionary process in Mammalian to determine melatonin dependent (independent) seasonal reproduction.

Title of the manuscript is fully adequate and informative about the scope of the research.

Abstract is well organized and very compressed info about the topic. Final conclusion is clear and could has an impact on future researches.

General comments

 Melatonin is synthesized in four steps (four enzymes) from which tryptophan hydroxylase (TPH) sets up the melatonin biosynthetic pathway. During the whole process there are two key conversion steps thus at first TPH is considered as the rate-limiting enzyme in serotonin synthesis.  TPH synthesis depends on the expression patterns of TPH1 and TPH2 genes. The TPH2 is confined to express in brain stem, whereas TPH1 expression products could be find in several tissues.

 Altogether 13 different melatonin related genes were analyzed in 28 mammalian species and 1 outgroup (zebrafish). Three of the investigated genes were found to be specialized during time. TPH1, TPH2 and IDO1 were further tested. They found a difference between the function of melatonin related proteins (TPH1, TPH2, IDO1, IDO2 and four CYP proteins). Site of the different AA is positioned to 442th, which is encoded by the 1325th nucleotide. The mutation resulted T/C changes. T is specific for seasonal reproductive pattern and it had been appeared 212.6 million years ago.

The research is very clear and well set up, the results are comprehensively discussed and concluded moderately.

However, before publication small changes advised to do detailed in the following section.

row 13 MLT row 55 MTNR.. MLT or MTN?

row 60 Petra et al, 1996 is Deprés-Brummer et al, 1996

row95 Mika &Lynch, 2016

row106 Thompson et al, 2018

row 318 Shu-Wen et al, 2016

row354 Keller et al, 2003

three references are not cited in the manuscript Bellivier et al, 2004; Dun-Xian et al, 2015 and Van de Peer et al, 2009

Author Response

Responses to the Reviewers’ Comments and Suggestions

Reviewer #1

Comments and Suggestions for Authors

row 13 MLT row 55 MTNR. MLT or MTN?

Reply: Thanks for your constructive comment. However, the “MLT” (line 13) in our paper means the melatonin and the “MTNR” (line 58) in the paper indicates the melatonin receptors, including MTNR1A and MTNR1B.

row 60 Petra et al, 1996 is Deprés-Brummer et al, 1996

row95 Mika &Lynch, 2016

row106 Thompson et al, 2018

row 318 Shu-Wen et al, 2016

row354 Keller et al, 2003

three references are not cited in the manuscript Bellivier et al, 2004; Dun-Xian et al, 2015 and Van de Peer et al, 2009

Reply: We apologize for the errors in references. We have accepted your valuable comment and modified the format and errors of references.

“Petra et al, 1996” is modified into [10] (line 61).

“Mika &Lynch, 2016” is modified into [40] (line 301).

Thompson et al, 2018 is modified into [45] (line 310).

Shu-Wen et al, 2016 is modified into [28] (line 225).

Keller et al, 2003 is modified into [29] (line 245).

Bellivier et al, 2004 has been deleted.

Dun-Xian et al, 2015 is modified into [13] (line 71).

Van de Peer et al, 2009 has been deleted.

Reviewer 2 Report

This manuscript describes allelic variations among 29 species in a variety of genes involved with melatonin synthesis and degradation as well as melatonin target site actions. The authors report on high allelic sequence and amino acid divergence in some genes (e.g. tryptophan hydroxylase, indoleamine dioxygenase), but not in others (e.g. cytochrome P450). In particular, they examined the variants of tryptophan hydroxylase 1 (TPH1) in which AA442 was found to be a proline in most non-seasonally breeding mammals and a leucine in most seasonally breeding mammals. Interestingly, the former TPH1 variant showed one-fifth of the in vitro enzymatic activity of the latter variant. Unfortunately, as this enzyme is not a rate-limiting step in melatonin synthesis, it is unclear what this finding means. The authors speculate a lot, but perhaps serotonin synthesis in the reticular activating system is impacted more by this gene variation than melatonin synthesis. An interesting discussion focuses on the evolution of seasonal reproduction as an outcome of positive selection for the TPH1 leucine variant in the face of the Cretaceous-Tertiary mass extinction event. However, the correlation of the TPH1 variant to the rise of various seasonally breeding mammals is just that – correlation, which does not prove causation. This is the major weakness of this otherwise good effort to understand the evolution of seasonal reproduction. Another weakness in this report is the language editing, beginning with the title “An ancient mutations…is consistent….”. Mutations are plural, hence it should read “Ancient mutations…..are consistent…”. The authors should seek out a native speaker to correct many grammatical problems in the text. Finally, the reference cited on page 3, line 99 (Nei et al., 2001) is missing in the References list.    

Author Response

Responses to the Reviewers’ Comments and Suggestions

Reviewer #2

Comments and Suggestions for Authors

the correlation of the TPH1 variant to the rise of various seasonally breeding mammals is just that – correlation, which does not prove causation.

Reply: Thanks for your constructive comment. We have modified our conclusion to “this polymorphism is consistent with the reproductive rhythm of mammals”, which is more consistent with the title and content of the article (line 378-380)

Moderate English changes required

Reply: We are grateful for your valuable comments to enrich our manuscript. Accordingly, the manuscript is edited by native English speaker and any mistake in the sentence structure and grammars are corrected in the revised version.

the reference cited on page 3, line 99 (Nei et al., 2001) is missing in the References list.

Reply: We apologize for the errors in references. We have accepted your valuable comment and found it is not important, so we deleted it.

Round 2

Reviewer 2 Report

The revised manuscript reads much better. However, in the abstract there are still some grammatical issues....

* lines 25/26 should be....." 89.5% of the amino acid of the non-seasonal reproducing mammals is was proline, while that of 88.9% of seasonal reproducing mammals is was leucine."

* line 30 should read...."212.6 million years ago, indicating that the evolution of the TPH1 gene was...."

Author Response

Comments and Suggestions for Authors

lines 25/26 should be " 89.5% of the amino acid of the non-seasonal reproducing mammals is was proline, while that of 88.9% of seasonal reproducing mammals is was leucine."

Reply: We are grateful for your valuable comments to enrich our manuscript. Accordingly, the mistake in the sentence structure and grammars are corrected in the revised version, and we have unified the description of seasonal reproducing mammals in our paper (lines 24, 25, 139, 140, 145, 146, 190, 191, 192, 228, 246, 249 and 267 in new version).

line 30 should read "212.6 million years ago, indicating that the evolution of the TPH1 gene was...."

Reply: Thanks for your comment, we have corrected it in the revised version (line 29).